# IMPDH Inhibition Decreases TERT Expression and Synergizes the Cytotoxic Effect of Chemotherapeutic Agents in Glioblastoma Cells

**DOI:** 10.3390/ijms25115992

**Published:** 2024-05-30

**Authors:** Xiaoqin Liu, Junying Wang, Laura J. Wu, Britni Trinh, Robert Y. L. Tsai

**Affiliations:** 1Institute of Biosciences and Technology, Texas A&M University Health Science Center, Houston, TX 77030, USA; xiaoqin.liu@exchange.tamu.edu (X.L.); aceg3389@gmail.com (J.W.); juichen814@gmail.com (L.J.W.); trinh354021@gmail.com (B.T.); 2Department of Translational Medical Sciences, College of Medicine, Texas A&M University Health Science Center, Houston, TX 77030, USA

**Keywords:** GTP, mycophenolic acid, synthetic lethality, telomerase, telomere

## Abstract

IMP dehydrogenase (IMPDH) inhibition has emerged as a new target therapy for glioblastoma multiforme (GBM), which remains one of the most refractory tumors to date. TCGA analyses revealed distinct expression profiles of IMPDH isoenzymes in various subtypes of GBM and low-grade glioma (LGG). To dissect the mechanism(s) underlying the anti-tumor effect of IMPDH inhibition in adult GBM, we investigated how mycophenolic acid (MPA, an IMPDH inhibitor) treatment affected key oncogenic drivers in glioblastoma cells. Our results showed that MPA decreased the expression of telomerase reverse transcriptase (TERT) in both U87 and U251 cells, and the expression of O^6^-methylguanine-DNA methyltransferase (MGMT) in U251 cells. In support, MPA treatment reduced the amount of telomere repeats in U87 and U251 cells. TERT downregulation by MPA was associated with a significant decrease in c-Myc (a TERT transcription activator) in U87 but not U251 cells, and a dose-dependent increase in p53 and CCCTC-binding factor (CTCF) (TERT repressors) in both U87 and U251 cells. In U251 cells, MPA displayed strong cytotoxic synergy with BCNU and moderate synergy with irinotecan, oxaliplatin, paclitaxel, or temozolomide (TMZ). In U87 cells, MPA displayed strong cytotoxic synergy with all except TMZ, acting primarily through the apoptotic pathway. Our work expands the mechanistic potential of IMPDH inhibition to TERT/telomere regulation and reveals a synthetic lethality between MPA and anti-GBM drugs.

## 1. Introduction

Glioblastoma multiforme (GBM) is the most common and aggressive type of primary brain tumor [1,2]. Conventional anti-glioblastoma therapies include surgery and radiation, but chemotherapy remains an integral part of the treatment, owing to the aggressive and infiltrative nature of GBM [3,4]. While many chemotherapies have been evaluated, their efficacy in treating GBM is restricted by tumor resistance and systemic toxicity [5,6]. The overall life expectancy for GBM patients remains unsatisfactory, with a five-year overall survival of only 6.8% [7] or an overall survival length of 12–15 months [8].

Recent advances in therapeutic intervention against GBM highlight several new developments with promising potential, including immunotherapies (e.g., bevacizumab) [9] and targeted therapies. Among the potential targeted therapies for GBM, a renewed interest in inosine 5′-monophosphate dehydrogenase (IMPDH) inhibition has emerged over the past few years. IMPDH is an important rate-limiting enzyme in the de novo purine synthesis pathway, which converts inosine monophosphate (IMP) to guanosine monophosphate (GMP) [10,11]. There are two IMPDH isoforms, IMPDH1 and IMPDH2, which share 84% protein similarity. Both isozymes are constitutively expressed in most tissues, with IMPDH1 more abundant in the spleen, retina, and leukocytes, and IMPDH2 more abundant in proliferating cells and neoplastic tissues. Cancer cells have a high metabolic demand for GMPs, which cannot be supplied solely through the purine salvage pathway. The reliance on GMP may explain the high expression level of IMPDH in cancer cells and their addiction to the de novo purine synthesis pathway. One recent study observed a large increase in IMPDH2 but not IMPDH1 in GBM compared to control brains and proposed IMPDH2 as the molecular link between GTP biosynthesis, the increased anabolic process of rRNA and tRNA, and tumor malignancy in GBM [12]. Another study showed that GBM cells adapted to temozolomide (TMZ) treatment by increasing the ARL13B–IMPDH2 interaction [13], suggesting that perturbation of this interaction may alter GBM sensitivity to TMZ.

While two studies published in 2018 raised the possibility of formulating new IMPDH2-inhibiting compounds with improved selectivity and potency [14,15], most IMPDH inhibition in research and the clinic has been achieved by mycophenolic acid (MPA), a noncompetitive reversible inhibitor of both IMPDH1 and IMPDH2. Clinically, MPA has been used as an immunosuppressant for organ transplantation and autoimmune diseases for many years [11,16]. Multiple studies have examined its anti-tumor effectiveness [17,18,19,20,21,22,23]. Early clinical trials fell short of asserting a justifiable use of MPA for cancer treatment, because (1) over 95% of the circulating MPA is metabolized by the liver to the inactive MPA glucuronide (MAG), which cannot be reverted back to MPA by most cancer cells [24]; (2) the therapeutic dose of MPA is limited by its gastrointestinal toxicity [25]; and (3) the acidic property of MPA may limit its penetration through the blood–brain barrier as a brain tumor therapy. Two studies have shown that MPA could synergistically enhance the anti-tumor efficacy of an Alb-kinase inhibitor (imatinib) [26] or several conventional chemotherapies [27,28].

Given the reemergence of interest in MPA (or its analogue) as a target therapy for GBM, a better understanding of how IMPDH inhibition intersects with key oncogenic drivers in adult GBM would provide important proof-of-mechanism data for MPA. To this end, we designed this study to determine the anti-GBM effect of IMPDH inhibition on pathways known to play key roles in driving adult GBM, including TERT, MGMT, EGFR, and PTEN/mTOR. MPA was used to inhibit IMPDH in U87 and U251 cells, because its cellular activity profile is better than that of investigational IMPDH2-specific inhibitors [15] and pan-inhibition of IMPDH avoids IMPDH1 compensation [12]. We discovered that MPA treatment decreased TERT expression and telomere repeats in both U87 and U251 cells. The decrease in TERT was associated with a significant decrease in c-Myc (a TERT transcription activator) in U87 cells and a dose-dependent increase in p53 and CTCF (TERT repressors) in both U87 and U251 cells. Furthermore, MPA synergistically enhanced the cytotoxic effect of chemotherapeutic agents in vitro via an apoptotic pathway. This study highlights the mechanistic and therapeutic repertoire of IMPDH inhibition in GBM treatment.

## 2. Results

### 2.1. Relative Abundance of IMPDH1 and IMPDH2 in GBM and LGG Samples

To survey the clinical relevance of IMPDH1 and IMPDH2 in glioma, we queried the gene expression data of human glioma samples from TCGA (https://genom-cancer.soe.ucsc.edu) (accessed on 15 June 2016) Gene expression levels were quantified by IlluminaHiSeq (RNA-seq percentile) in 516 LGG and 172 GBM samples or by Affymetrix chips in 529 GBM samples of different transcriptional features. Based on the RNA-seq data, GBM samples expressed more IMPDH1 and less IMPDH2 compared to LGG (Figure 1A). Based on the Affymetrix data, IMPDH1 was most abundant in the mesenchymal and least in the neural subtype, whereas IMPDH2 was most abundant in the proneural subtype and least in the neural subtype (Figure 1(B1)). TCGA classifications also include localized, progression, and recurrent GBMs, referring to tumors that are newly diagnosed (localized), advanced/worsening over time despite initial treatment (progression), or recurrent after a period of remission following initial therapy (recurrent). IMPDH1 showed no difference, whereas IMPDH2 was more abundantly expressed in localized GBMs than progressive or recurrent GBMs (Figure 1(B2)). GBMs with the CpG island methylation phenotype (G-CIMP) expressed more IMPDH2 and less IMPDH1 compared to those without (Figure 1(B3)). Among LGG samples, IMPDH1 was expressed more abundantly in WHO-defined tumor grade 3 (G3) than G2, whereas IMPDH2 showed no difference (Figure 1(C1)). In terms of outcome, IMPDH1 showed no difference between LGGs of different prognosis, whereas IMPDH2 was more abundantly expressed in LGG with a better prognosis than those with a poor prognosis (Figure 1(C2)). Consistent with the latter finding, IMPDH2 was expressed more abundantly in LGG with a longer relapse-free survival (RFS) (Figure 1(C3)), a longer time to death (Figure 1(C4)), and a longer time to new tumors (Figure 1(C5)), whereas IMPDH1 showed no statistical difference in expression between LGGs of good vs. poor outcomes. These TCGA results show that IMPDH1 and IMPDH2 display mild but statistically significant differences in their relative abundance in GBM and LGG with different transcriptomic, methylation, and prognostic features. Finally, one should note that the NIH’s TCGA database classified grade II and III astrocytoma and oligodendroglioma as LGG [29] and grade IV glioma as HGG. This classification does not completely conform to the most recent WHO CNS5 standards in 2021 [30].

### 2.2. MPA Effect on the Expression of MGMT, TERT, EGFR, and S6k1 in U87 and U251 Cells

To explore how IMPDH inhibition impacted GBM tumorigenesis at the molecular level, we determined the effect of MPA on several oncogenic pathways known to play critical roles in adult primary GBM [31]. Gene-specific primers were designed to measure the expression of MGMT, TERT, EGFR, S6K1 (a downstream target of the PTEN/mTOR pathway), and two gene products of CDKN2A (i.e., p16^INK4a^ and p14^ARF^) by qRT-PCR. Gene product amounts were compared to two non-nucleolar, non-ribosomal reference genes, high mobility group nucleosome binding domain 1 (HMGN1) and glyceraldehyde-3-phosphate dehydrogenase (GAPDH). Compared to vehicle treatment, MPA treatment (6 μM or 24 μM for 24 h) increased MGMT expression by 2 to 2.5-fold in U87 cells (Figure 2(A1,A2)), but decreased its expression by 32–35% in U251 cells (Figure 2(B1,B2)). In contrast, MPA treatment showed a significant effect in downregulating TERT expression in both U87 cells (by 80–90%) and U251 cells (by 50–60%) (Figure 2). Beside these two findings, MPA had no effect on the expression of EGFR or S6K1 in either U87 or U251 cells (Figure 2), and the expression levels of p16^INK4a^ and p14^ARF^ were too low to be detected by qRT-PCR in either cell lines with or without MPA treatment. Our results show that, among the key oncogenic pathways in adult primary GBM, MPA exerts a specific and significant effect in downregulating TERT expression in both U87 and U251 cells, and that the TERT-inhibitory effect of MPA is stronger in U87 cells than in U251 cells.

### 2.3. Effect of MPA Treatment on Telomere Length in U87 and U251 Cells

To determine whether MPA-induced TERT downregulation resulted in telomere length shortening, U87 and U251 cells were treated with vehicle or MPA (6 μM or 24 μM) for 24 h, replaced with fresh media, and grown in culture for an additional 3 days (3 d), 7 d, or 10 d. Genomic DNA samples were collected at each timepoint and quantified for their telomere length using a qPCR-based telomere length assay. First, we validated that the telomere primer pair (Tel-G and Tel-C) and the reference gene primer pair (36B4u and 36B4d) measured the DNA template amounts in a quantitatively linear fashion within the range of 10–160 ng template DNA in 20 μL reaction (Figure 3A). Supporting a functional consequence of MPA-induced TERT downregulation, our results showed that MPA treatment reduced the amount of telomere repeats at 3 d, 7 d, and 10 d after the treatment, and that 24 μM MPA displayed a stronger effect than 6 μM MPA in U251 cells (Figure 3(B1)). MPA treatment also reduced the amount of telomere repeats in U87 cells but only at 10 d after the treatment (Figure 3(B2)). The baseline levels of telomere repeats were 30–40% higher in U251 cells compared to U87 cells (Figure 3C).

### 2.4. Transcriptional Regulation of TERT Activators and Repressors by MPA Treatment

TERT transcription is regulated by multiple transcription factors (TFs) binding to its promoter. Canonical positive regulators of TERT include c-MYC, Sp1, NF-kB RelA (p65), signal transducer and activator of transcription (STAT) family proteins (e.g., STAT3), adipocyte protein 2 family (e.g., TFAP2A and TFAP2B), Goosecoid homebox protein (GSC), SET and MYND Domain Containing 3 (SMYD3), and NRAS [32,33,34,35,36,37,38,39,40,41,42,43,44,45]. GA-binding proteins A (GABPA) and B1 (GABPB1) have been shown to directly activate a mutant TERT promoter (−124C>T) [46,47,48,49]. This mutation was found in U87 and U251 cells in monoallelic or homozygous form, respectively [50,51]. TERTp also contains multiple binding sites for transcriptional repressors, including mitotic arrest deficient 1 (MAD1), p53, CCCTC-binding factor (CTCF), and Wilms’ tumor 1 (WT1) [36,52,53,54,55,56,57,58,59]. To look for potential upstream mechanisms by which IMPDH inhibition downregulates TERT transcription, we studied how MPA treatment affected the levels of all those factors in U87 and U251 cells, except for those whose expressions are too low to be reliably quantified by qRT-PCR (e.g., GSC, WT1, TFAP2B). For positive TERT regulators, we showed that MPA significantly decreased the expression of c-MYC in U87 cells but not in U251 cells (Figure 4A), increased the expression of STAT3 in both U87 and U251 cells (Figure 4B), and had no effect on the expression of TFAP2A (Figure 4C), GABPA (Figure 4D), Sp1, p65, SMYD3, or NRAS. For negative TERT regulators, we showed that MPA increased the expression of p53 and CTCF in a dose-dependent manner in both U87 and U251 cells (Figure 4E,F). These data reveal that MPA-mediated TERT downregulation is associated with a decrease in c-MYC activator in U87 cells and increases in p53 and CTCF repressors in both U87 and U251 cells.

### 2.5. Synergy between MPA and BCNU and Other Drugs

MGMT plays a major role in conferring cancer resistance to alkylating agents. Given the inhibitory effect of MPA on MGMT in U251, we hypothesized that MPA might synergize the anti-GBM activity of BCNU in U251 but not in U87 cells. To test this idea, we analyzed the drug interaction between MPA and BCNU, an alkylating agent used in GBM treatment, by their combination indices (CIs). The IC_50_ concentrations of MPA and BCNU in U87 and U251 cells were determined by their dose response curves based on the MTT assay. Five different ratios of mixture were tested, including 4-to-1, 2-to-1, 1-to-1, 1-to-2, and 1-to-4, with the 1-to-1 ratio representing each drug at half of their respective IC_50_ concentrations. The amounts of cells surviving the treatment were measured by the MTT assay. CIs were calculated using Isobologram software version 2.0. Strong, moderate, and mild synergism, and additive drug interactions were defined by CI values in the range of 0.1–0.3, 0.3–0.7, 0.7–0.85, and 0.85–1.0, respectively. Our results showed that MPA worked in strong synergy with BCNU in U251 cells, with the lowest CI value (0.14) at the 1-to-1 BCNU-to-MPA ratio of combination, followed by the 2:1, 4:1, 1:4, and 1:2 BCNU-to-MPA combinations (Figure 5A, right). This result is consistent with the MPA effect in downregulating MGMT in U251 cells. Interestingly, MPA also worked in strong synergy with BCNU in U87 cells, with the lowest CI value (0.20) at the 1-to-1 BCNU-to-MPA ratio (Figure 5A, left), suggesting that other MGMT-independent mechanisms might be involved.

Given that MPA downregulated TERT in both U87 and U251 cells, we hypothesized that the synergizing effect of MPA with multiple chemotherapeutic agents might be mediated via TERT downregulation. Indeed, we found that in U87 cells, MPA also worked in strong synergy with IRI (the lowest CI value of 0.11 observed at the 1-to-1 IRI-to-MPA ratio of combination), OXP (the lowest CI value of 0.29 observed at the 4-to-1 OXP-to-MPA ratio of combination), and Taxol (the lowest CI value of 0.23 observed at the 1-to-1 Taxol-to-MPA ratio of combination) (Figure 5B–D, left panels), and in moderate synergy with TMZ (the lowest CI value of 0.37 observed at the 1-to-1 TMZ-to-MPA ratio of combination) (Figure 5E, left panel). In contrast, MPA exerted only moderate synergy with IRI (the lowest CI value of 0.43 observed at the 1-to-1 IRI-to-MPA ratio of combination), OXP (the lowest CI value of 0.38 observed at the 4-to-1 OXP-to-MPA ratio of combination), Taxol (the lowest CI value of 0.45 observed at the 1-to-1 Taxol-to-MPA ratio of combination), and TMZ (the lowest CI value of 0.53 observed at the 1-to-1 TMZ-to-MPA ratio of combination) in U251 cells (Figure 5B–E, right panels). The relative synergy between MPA and multiple chemotherapeutic agents in U87 vs. U251 cells was consistent from experiment to experiment (Figure 5F). Next, we compared the synergy of combining two of the three anti-GBM drugs (e.g., TMZ, BCNU, and IRI) versus the synergy of their individual combination with MPA. Our results showed that the TMZ–IRI combination displayed moderate cytotoxic synergy in both U87 cells (lowest CI = 0.54 at the 1-to-2 ratio of combination) and U251 cells (lowest CI = 0.68 at the 2-to-1 ratio of combination) (Figure 6(A1,A2)). The TMZ–BCNU combination had no synergy in U87 cells (lowest CI = 0.96 at the 1-to-1 ratio of combination) and moderate synergy in U251 cells (lowest CI = 0.56 at the 4-to-1 ratio of combination) (Figure 6(B1,B2)). The IRI–BCNU combination exhibited moderate synergy in both U87 cells (lowest CI = 0.58 at the 2-to-1 ratio of combination) and U251 cells (lowest CI = 0.41 at the 2-to-1 ratio of combination) (Figure 6(C1,C2)). Together, our data suggest that downregulation of TERT (in U87 and U251 cells) and BCNU (in U251 cells) by MPA treatment may synergize the cytotoxic effect of chemotherapeutic agents in GBM cells.

### 2.6. Combining MPA with BCNU, OXP, Taxol, or TMZ Increases Apoptosis in U251 Cells

To establish the cytotoxic synergy between MPA and chemotherapeutic agents (e.g., BCNU, IRI, OXP, Taxol, or TMZ), U251 cells were treated with single agents or two-drug combinations for 24 h and measured for their percentage of apoptotic and necrotic cells. To better show the synergistic effect of MPA-combined treatment, cells were treated with single agents at 1/10th of their respective IC_50_ concentrations or two-drug combinations at 1/20th of their respective IC_50_ concentrations. Cytotoxicity was determined using the Biotium Apoptosis and Necrosis Quantitation Kit Plus, where apoptotic cells were labeled with CF488A-conjugated Annexin V and necrotic cells were determined by their EthD-III+/Annexin V-signals. Flow cytometry demonstrated that U251 cells treated with MPA alone at 12 μM did not show more apoptotic or necrotic cells than control-treated cells (Figure 7(A1,B1)). Notably, cells treated with BCNU (2.5 μM) plus MPA (6 μM) showed a significant increase in apoptotic cells (22.7%) compared to those treated with BCNU alone at 5 μM (12.6%) (Figure 7(A2,B2)). Cells treated with OXP (0.5 μM) plus MPA (6 μM) also showed an increase in apoptotic cells (21.9%) compared to those treated with OXP alone at 1 μM (16.3%) (Figure 7(A4,B4)). Similarly, MPA addition (6 μM) increased the percentage of apoptotic cells in the combined agent-treated groups compared to the single agent-treated groups, including those treated with Taxol (18.9% vs. 9.1%) (Figure 7(A5,B5)) or TMZ (19.1% vs. 10.4%) (Figure 7(A6,B6)). The percentage of necrotic cells was increased by MPA addition only in those groups treated with Taxol (9.3% vs. 4.3%) (Figure 7(A5,B5)). Finally, MPA addition showed no effect on increasing the percentage of either apoptotic or necrotic cells treated with IRI, suggesting a non-cytotoxic mechanism for the IRI–MPA combination (Figure 7(A3,B3)). These findings confirm that MPA in combination with BCNU, OXP, Taxol, or TMZ creates cytotoxic synergy in GBM cells, and that this cytotoxic effect primarily represents an increase in apoptosis.

## 3. Discussion

The malignant growth and drug resistance of GBM pose the two most challenging issues in brain tumor research and treatment that underscore its high mortality rate. Recent reports on the addiction of GBM to IMPDH [12,60] triggered our interest in further dissecting the anti-tumor effect of IMPDH inhibition on several key oncogenic drivers in adult primary GBM [31], including increased expression of MGMT and TERT [61,62], EGFR amplification [63], PTEN deletion/mTOR dysregulation [64], and homozygous CDKN2A deletion [65].

### 3.1. IMPDH Inhibition Downregulates TERT Expression

A new discovery made in this study was the effect of IMPDH inhibition in downregulating TERT expression in U251 and U87 cells. By comparison, MPA decreased TERT expression more in U87 cells than in U251 cells, which showed a higher baseline expression of TERT than U87 cells. Consistent with the overall importance of telomere in dividing cells, MPA displayed a strong synergy in U87 cells and a moderate synergy in U251 cells when combined with a wide range of chemotherapeutic agents. In addition to TERT regulation, we also noted a cell-type-dependent effect of MPA on decreasing the expression of MGMT in U251 cells and increasing its expression in U87 cells. MGMT was shown to play a major role in conferring cancer resistance to BCNU and other alkylating agents [66,67]. BCNU creates two types of DNA lesions: chloroethyl adducts at O^6^-guanine and interstrand cross-links [68]. As repair of chloroethyl adducts is mediated by MGMT, epigenetic silencing of MGMT via promoter methylation is associated with a favorable response to alkylating agents in glioblastoma patients [69]. MPA-induced MGMT downregulation of U251 cells provides a molecular explanation for MPA’s strong synergy with BCNU but only moderate synergy with other drugs. Lastly, MPA showed no effect on the expression of EGFR or S6K1 (a downstream target of the PTEN/mTOR pathway), and the expressions of the two tumor suppressor gene products of CDKN2A (p16^INK4a^ and p14^ARF^) were too low to be detected, with or without MPA treatment, in either U87 or U251 cells. The specific effect of MPA on TERT but not the others (e.g., EGFR, S6k1) indicates that our findings were unlikely to have been caused by global defects in DNA replication or RNA transcription machinery, which should affect multiple targets, including EGFR and S6k1, as well as the two internal reference genes, HMGN1 and GAPDH.

TERT transcription is regulated by multiple TFs with binding sites on the TERT promoter. Analyses of 12 positive regulators and 4 negative regulators for TERT transcription showed that MPA treatment caused a significant decrease in c-MYC in U87 cells and a dose-dependent increase in p53 and CTCF in both U87 and U251 cells. These findings suggest that the significant downregulation of TERT by MPA in U87 cells may have been due to combined changes in c-MYC, p53, and CTCF, whereas its moderate decrease in U251 cells was caused by changes in p53 and CTCF but not c-MYC. Finally, STAT3 upregulation of MPA-treated U87 and U251 cells may represent a compensatory mechanism in response to TERT downregulation. We speculate that MPA’s effect on TERT downregulation may provide a mechanism for suppressing GBM growth, in parallel with that of GTP biosynthesis and the anabolic process of rRNA and tRNA [12].

A qRT-PCR assay detects gene expression changes at the transcript level. It is more sensitive and quantitatively feasible compared to Western blotting of a large-sized protein such as TERT (120 kDa). To connect gene expression findings to function, we provided data on telomere length. U87 and U251 cells are both TERT+ cells and depend on a TERT promoter mutation (−124C>T) for long-term (30-day) proliferation [70]. The alternative lengthening of telomeres (ALT) mechanism in those cells plays more of a compensatory role than a driver role. Consistently with its effect on downregulating TERT expression, MPA treatment reduced the amount of telomere repeats in U87 and U251 cells over time. Even though the decrease in TERT by MPA was more obvious in U87 cells than in U251 cells, the decrease in telomere repeats appeared at earlier timepoints in U251 compared to U87 cells. This discrepancy may be explained by (1) a longer double time of U87 cells (39–72 h) compared to that of U251 (23–24 h) [71,72], and/or (2) the potential lesser reliance of TERT in telomere maintenance in U87 cells than U251, as suggested by the lower baseline level of TERT in U87 cells compared to U251 cells. To minimize the cell death effect of MPA in confounding its effect on telomere length maintenance, cells were transiently (24 h) treated with low-dose MPA at 6 μM and 24 μM, corresponding to 1/20th and 1/5th of its IC_50_ concentration, respectively. To determine whether the MPA effect on TERT expression was sustained throughout the 10 days, qRT-PCR assays performed on parallel samples confirmed that the TERT-inhibitory effect of MPA was sustained throughout the time window of the experiment. The extent of telomere repeat loss (~30% in 10 days) in our study was greater than the average attrition rate per replication cycle in normal cells (30–200 bp) with the estimated doubling number for U87 cells (3–6 times in 10 days) and U251 cells (10 times in 10 days). We, therefore, reasoned that the loss of telomere repeats in MPA-treated cells was not only caused by the end-replication problem but also compounded by oxidative damage and/or a shortage of guanine nucleotides for repair/synthesis.

### 3.2. Cytotoxic Synergy between MPA and Anti-GBM Agents

Another finding of this study was the cytotoxic synergy of MPA in combination with anti-GBM agents. For almost all the drugs we tested, the relative synergy of MPA in U87 vs. U251 cells paralleled the MPA effect on TERT expression, except for BCNU. In U251 cells, MPA displayed a strong synergy with BCNU but only moderate synergy with other chemotherapeutic agents, which might have been due to its effect in downregulating MGMT expression in U251 cells. A previous paper reported that the MPA synergy may differ between cell lines of different tumor types [27]. Our study showed that U251 and U87 cells, even though both were derived from GBM tumors, may exhibit different behaviors in response to MPA treatment. Since their establishment by Ponten et al. in 1975, U251 and U87 cells have been extensively used in GBM research [73]. Differences have been shown in their histopathology, genetics, and epigenetics. For example, U251 cells overexpress EGFR and harbor a R273H mutation in their p53 gene, resulting in the abolishment of its DNA binding activity, whereas U87 cells express wildtype EGFR and p53. U251 cells are also more enriched in CD133^+^ glioma stem cells than U87 cells [74]. Those differences may account for the cell-line-dependent variation in MPA synergy as well as its underlying mechanisms. BCNU is used clinically in a local release formulation (Gliadel Wafer) as a peri-operative treatment of GBM. This localized drug delivery formulation and our discovery of the BCNU–MPA combination may form the basis of a new design for GBM treatment by combining with a computer-aided 3D printing design technology [28]. A topical delivery design of BCNU–MPA may obviate potential issues related to the pharmacokinetics and toxicities of systemically administered MPA, such as its rapid metabolism, dose-limiting gastrointestinal toxicity, and CNS bioavailability and pharmacokinetics (e.g., BBB penetration). A trade-off in adding an IMPDH inhibitor to GBM treatment is its immunosuppressive effect, which may diminish the body’s immune response against tumor cells or their response to anti-PD-1/PD-L1 immunotherapies.

### 3.3. Limitations and Challenges Ahead

The focus of this study was on the molecular mechanism underlying the anti-GBM effect of MPA. Our strategy was to explore the intersection between IMPDH inhibition and pathways that have been shown to play important roles in the oncogenesis of adult GBM. While this approach offered the advantage of connecting the MPA effect to those of established pathophysiological importance, we cannot exclude the possibility that other yet-to-be-discovered pathways may also be involved. To address this issue, high-throughput multi-omics screens could be used to make new unbiased discoveries, complementary to the target-selective approach used in this study. Conversely, the downside of these whole-genome approaches is the difficulty in establishing the functional and clinical relevance of their output targets. Finally, it is worth noting that we focused our study on pathways related to adult primary GBM and not on pathways involved in (1) low-grade glioma or secondary high-grade glioma, such as mutations on isocitrate dehydrogenase 1 (IDH1) and IDH2 [75] and loss of alpha thalassemia/mental retardation syndrome X-linked (ATRX) [76]; or (2) pediatric glioma, such as mutations on B-Raf V600E mutation [77] and H3.1/H3.3 K27M mutation. Of note, ALT+ tumors constitute a much smaller percentage of adult GBM (16%) compared to pediatric GBM (39%), grade II astrocytoma (55%), or grade III astrocytoma (65%) [78].

The increased sensitivity of GBM cells to the MPA-combined therapy raises new possibilities for developing anti-GBM therapies. For any new discovery, many proof-of-mechanism and proof-of-principle validation studies have to be carried out in cell models, animal models, and FDA-approved randomized clinical trials before its final application in the clinic. All preclinical models have their strength(s) and weakness(es). While primary cells and patient-derived xenograft (PDX) models are less deviant from the original tumors, they suffer a higher degree of heterogeneity within the tumor or between tumors from different patients, due to their unique clinicodemographic characters, genetic makeup, and underlying carcinogenic events. Those intra-tumor and inter-tumor differences may create issues of comparability and reproducibility between studies. Furthermore, primary cells and PDX models are also subject to culture or transplant-related selection and changes to some extent. Established cell line models, such as U251 and U87 cells, have been widely used by the GBM research community for several decades. While they are more deviant from primary tumors compared to primary cell culture or PDX models, established cell lines are well characterized at the histopathological, molecular, and genetic levels, and display relatively homogeneous and stable phenotypes and molecular features within limited passages. As a result, they have been extensively used for discovery and mechanistic work. Our study reports a new mechanism by which MPA negatively controls GBM cell proliferation. Whether it can ultimately be applied as a therapeutic agent for GBM still requires further studies on its efficacy, pharmacokinetics, and toxicity, using preclinical rodent and non-rodent animal models, as well as clinical trials.

## 4. Materials and Methods

### 4.1. TCGA Analysis

The TCGA gene expression datasets for GBM (RNA-seq and Affymetrix) and low-grade glioma (LGG) (RNA-seq) were downloaded from UC Santa Cruz Cancer Browser and analyzed as described previously [79,80,81]. The LGG RNA-seq dataset contains 516 samples of primary astrocytoma, oligodendroglioma, and oligoastrocytoma of grade 2 or 3. The GBM RNA-seq dataset contains 172 samples. The GBM Affymetrix dataset contains 529 samples. Correlation between categorical clinical parameters and gene expression levels was analyzed by *t*-test statistics.

### 4.2. Cell Culture

Human glioma U87 and U251 cell lines were obtained from American Type Culture Collection (ATCC) (Manassas, VA, USA) and MD Anderson Cancer Center (Houston, TX, USA) and confirmed in their identities by short tandem repeat (STR) fingerprinting on 4 August 2016 and 14 June 2016, respectively. Cells were maintained in monolayer culture in DMEM medium, supplemented with 10% FBS (Hyclone), penicillin (50 IU/mL), streptomycin (50 μg/mL), and glutamine (1%) at 37 °C in humidified incubators with 5% CO_2_ [82].

### 4.3. Quantitative (Real-Time) RT-PCR Assay

U87 and U251 cells were treated with 0 μM, 6 μM (1/20th of IC_50_), and 24 μM of MPA, and collected after 24 h. Total RNAs were extracted using the TRIzol^TM^ reagent and reverse transcribed into 1st-strand cDNA using M-MLV reverse transcriptase, as described previously [83,84]. For quantitative PCR reactions, the ΔC(t) values between target genes and reference genes, i.e., HMGN1 and glyceraldehyde-3-phosphate dehydrogenase (GAPDH), were determined using the MyiQ single-color real-time PCR detection system (BUN9740RAD) and supermix SYBR green reagent (170-8882) (Bio-Rad, Hercules, CA, USA). Tm was set at 60 °C for all reactions. The ΔΔC(t) values were calculated by comparing MPA-treated samples to non-treated samples (set as 100%). All results represent 6–8 biological replicates with 2–3 technical repeats each. Primer sequences are listed below (Table 1).

### 4.4. qPCR-Based Telomere Length Assay

Telomere length was measured by a quantitative PCR (qPCR) method. Genomic DNAs were extracted from cells or tissues using a NucleoSpin Tissue kit (TaKaRa Bio, #740952, San Jose, CA, USA). Telomere length was quantified by the relative ratio of copy numbers between telomere repeats and a single copy gene. qPCR reactions were prepared in 20 μL containing 100 ng of genomic DNA, 1.25 μM of Tel-G and Tel-C primer (each) in SYBR Green PCR Master Mix (BioRad, 170-8884BUN, Fort Worth, TX, USA). Reactions were run in a MyiQ single-color real-time PCR detection system with three cycles of 95 °C for 10 min, 94 °C for 16 s, and 49 °C for 16 s, followed by 40 cycles of 94 °C for 16 s, 62 °C for 16 s, and 76 °C for 20 s. Primer sequences for telomere repeats were Tel-G: 5′-ACACTAAG(GTTTGG)_4_GTTAGTGT-3′ and Tel-C: 5′-TGTTAGG(TATCCC)_5_TAACA-3′. Primer sequences for single copy gene (RPLP0) were human 36B4d (5′-CCCATTCTATCATCAACGGGTACAA-3′, sense), human 36B4u (5′-CAGCAAGTGGGAAGGTGTAATCC-3′, anti-sense).

### 4.5. Thiazolyl Blue Tetrazolium Bromide (MTT) Assay

Cells were plated in 96-well plates at 2000 cells per well one day before receiving the drug treatment. Drugs tested in this study included MPA (Sigma, M-5255, Rockville, MD, USA), BCNU (carmustine) (Sigma, C-0400), irinotecan (IRI) (Sigma, I-1406), oxaliplatin (OXP) (Sigma, O-9512), Taxol (Cayman Chemical, 10461-25, Ann Arbor, MI, USA), and TMZ (Sigma, F-2577). Drug concentrations were specified in individual experiments. After 3 days of treatment, the remaining cells were quantified using an MTT assay. The MTT assay was performed by adding the MTT solution (10 μL, 5 mg/mL) (Sigma, M-5655) to each well and incubating for 4 h to allow MTT to be metabolized to formazan. Formazan was then dissolved in DMSO and measured by spectrophotometer at 560 nm.

### 4.6. Combination Index (CI) and Isobologram

The tumor growth-inhibitory curves of individual drugs were first measured by the MTT assay over a range of five concentrations with three-fold increment [27,28]. The half-inhibitory concentrations (IC_50_) were calculated by the Calcusyn program version 2.0 (Biosoft, Ferguson, MO, USA) (Table 2). The combined effects were measured by mixing two drugs at a 4-to-1, 2-to-1, 1-to-1, 1-to-2, or 1-to-4 ratio according to their respective IC_50_ concentrations. Each combination was tested for its tumor growth-inhibitory effect over a range of five different dosages. The ED_50_ CI values for each condition were calculated using the Calcusyn program and presented as the averages of 3–5 biological replicates.

### 4.7. Apoptosis and Necrosis Measurement

U251 cells were plated at 3 × 10^5^ cells per well density on 6-well plates and treated with single agent at 1/10th of its IC_50_ concentration or with a two-drug combination at 1/20th of their respective IC_50_ concentrations for 24 h. One day later, unfixed cells were trypsinized and dissociated into single cell suspension. Apoptotic and necrotic cells were sequentially stained with CF488A-conjugated Annexin V and ethidium homodimer III (EthD-III), respectively, using a Biotium Apoptosis and Necrosis Quantitation Kit Plus (Biotium, #30065, Fremont, CA, USA). Samples were analyzed on a BioRad ZE5 cell analyzer equipped with 5 lasers (UV 305, violet 405, blue 488, Y/G 561, and red 640 nm excitation wavelengths) and 27 fluorescence detectors [85]. Apoptosis was measured by labeling with Annexin-CF488a and detected using the 488 nm laser on the FITC filter (525/35). Necrosis was measured by labeling with ethidium homodimer III and detected using the 593/52 filter off the 488 nm laser. Data were analyzed by FlowJo 10.8.0 software. Apoptotic and necrotic cells were defined as those showing CF488A+ and CF488A−/EthD-III+ signals, respectively. Final data represent the average of four to six independently performed experiments.

## 5. Conclusions

This study reports new mechanisms underlying the anti-GBM effect of IMPDH inhibition via TERT downregulation and a strong cytotoxic synergy between MPA and multiple anti-GBM reagents via the apoptotic pathway. Among the five chemotherapeutic agents tested, BCNU displayed the strongest synergy with MPA. TERT downregulation by MPA treatment was associated with a decrease in c-MYC (a TERT activator) in U87 cells and dose-dependent increases in p53 and CTCF (TERT repressors) in U87 and U251 cells. This work expands our mechanistic understanding of IMPDH inhibition in suppressing GBM.

## Figures and Tables

**Figure 1 ijms-25-05992-f001:**
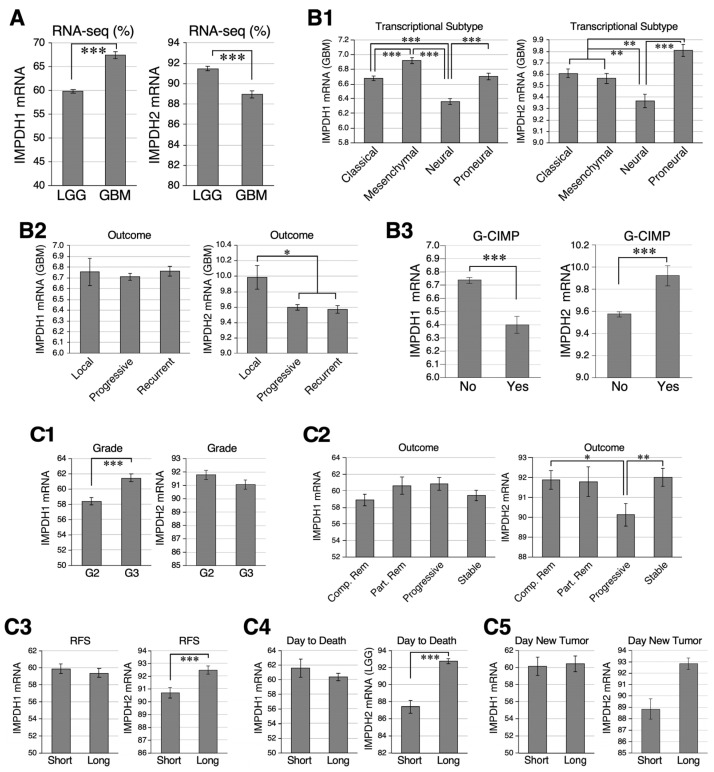
The Cancer Genome Atlas (TCGA) analysis of IMP dehydrogenase 1 (IMPDH1) and IMPDH2 expression levels in human glioblastoma multiforme (GBM) and low-grade glioma (LGG) samples. (**A**) IMPDH1/2 expression (RNA-seq) in LGG vs. GBM. (**B**) IMPDH1/2 expression (Affymetrix) in GBMs with different transcriptomic profiles (**B1**), progression outcomes (**B2**), and CpG Island Methylation (G-CIMP) phenotypes (**B3**). (**C**) IMPDH1/2 expression (RNA-seq) in LGGs with different WHO-defined tumor grades (**C1**), progression outcomes (**C2**), relapse-free survival (RFS) time (**C3**), day to death (**C4**), and day to new tumors (**C5**). Bars show means ± s.e.m. with asterisk(s) indicating *p* values, i.e., * (<0.05), ** (<0.01), and *** (<0.001). Note: In this TCGA analysis, LGGs include grade II and III astrocytoma and oligodendroglioma.

**Figure 2 ijms-25-05992-f002:**
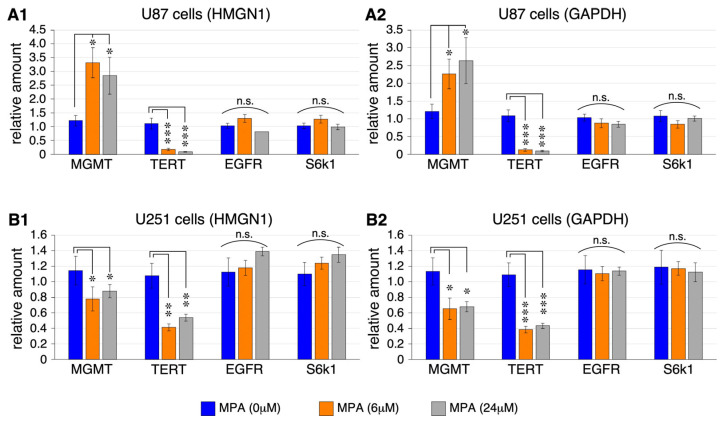
Effect of IMPDH inhibition on the expression of O^6^-methylguanine-DNA methyltransferase (MGMT), telomerase reverse transcriptase (TERT), epidermal growth factor receptor (EGFR), and S6K1 in U87 and U251 cells. Expression levels of MGMT, TERT, EGFR, and S6K1 transcripts were measured by qRT-PCR assay in U87 cells (**A**) and U251 cells (**B**). Cells were treated with vehicle (blue bars), 6 μM mycophenolic acid (MPA) (orange bars), or 24 μM MPA (grey bars) for 24 h. Amounts were normalized to two internal controls, HMGN1 (**A1**,**B1**) or GAPDH (**A2**,**B2**). Graphs show means (±s.e.m) of four biological replicates and 2 technical repeats each. Abbreviations: n.s., not statistically significant. *p*-values: *, <0.05; **, <0.01; ***, <0.001.

**Figure 3 ijms-25-05992-f003:**
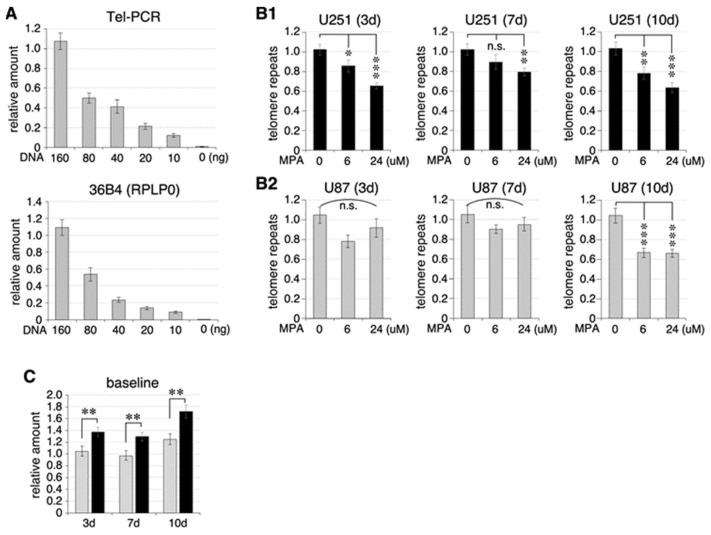
MPA treatment decreases the amounts of telomere repeats in U87 and U251 cells. (**A**) Validation of linear range detection of telomere repeats (Tel-PCR) and a reference gene (36B4) using qPCR assays. (**B**) U251 cells (black bars, (**B1**)) and U87 cells (grey bars, (**B2**)) were treated with vehicle or MPA (6 or 24 μM) for 24 h; grown in culture for an additional 3, 7, and 10 days; and measured for telomere length by the qPCR-based assay. The amount of telomere repeats was normalized to the amount of a single-copy house-keeping gene (36B4) in the same sample and then compared to vehicle-treated samples maintained in culture for the same time period. (**C**) Comparison of baseline levels of telomere repeats in U87 (grey) and U251 (black) cells collected at 3 d, 7 d, and 10 d timepoints. Bars show means ± s.e.m. of six biological replicates and three technical repeats each. *p*-values: *, <0.05; **, <0.01; ***, <0.001; n.s., no significant difference.

**Figure 4 ijms-25-05992-f004:**
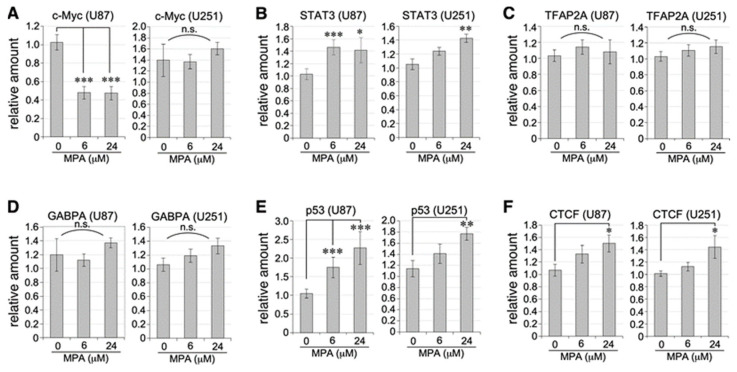
MPA effect on the transcriptional regulation of TERT. Expression levels of TERT transcriptional activators, including c-MYC (**A**), STAT3 (**B**), TFAP2A (**C**), and GABPA (**D**), and TERT transcriptional repressor, including p53 (**E**) and CTCF (**F**), in U87 and U251 cells treated with vehicle or MPA (6 or 24 μM) for 24 h. Graphs show results compared to HMGN1 as the internal reference. Bars show means ± s.e.m. of eight biological replicates and two technical repeats each. *p*-values: *, <0.05; **, <0.01; ***, <0.001; n.s., no significant difference.

**Figure 5 ijms-25-05992-f005:**
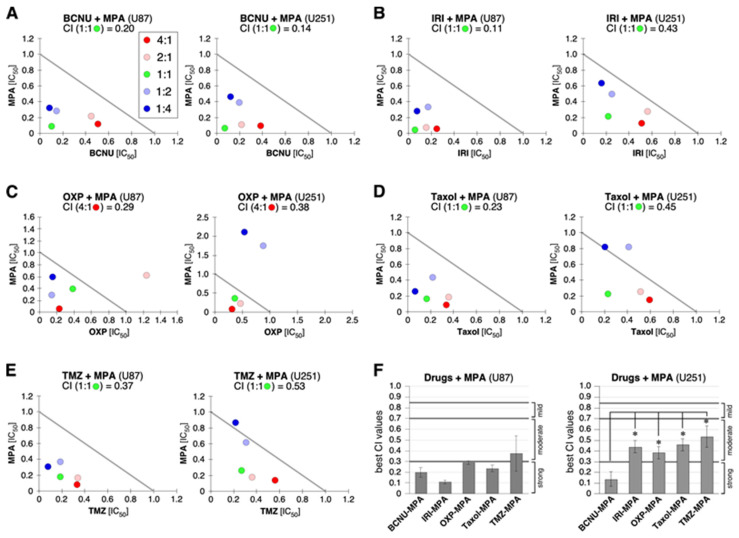
Synergy between MPA and BCNU (carmustine), irinotecan (IRI), oxaliplatin (OXP), paclitaxel (Taxol), or temozolomide (TMZ) in U87 and U251 cells. Synthetic lethality between MPA and BCNU (**A**), IRI (**B**), OXP (**C**), Taxol (**D**), or TMZ (**E**) was determined in U87 cells (left) and U251 cells (right) by the MTT assay. Combination indices (CIs) were calculated using the Isobologram (Calcusyn) program. Ratios show the relative amounts of individual drugs to MPA, with (1:1) representing drugs at half of their respective IC_50_ concentrations. The ratios showing the strongest synergy for each pair are listed on top. (**F**) Statistical analyses of the best CI values for each drug combination. Graphs show means (±s.e.m) of 3–4 biological replicates. *p*-values: *, <0.05.

**Figure 6 ijms-25-05992-f006:**
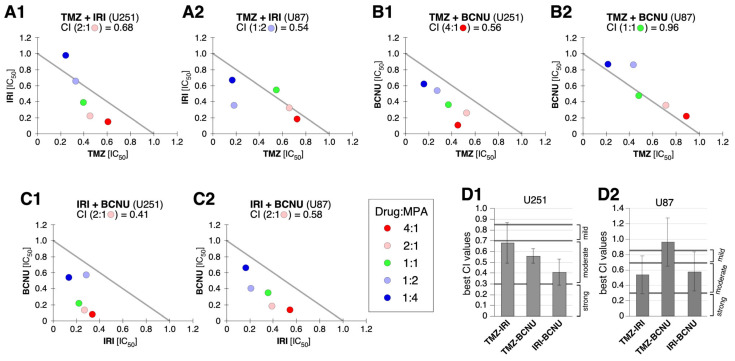
Two-drug combinations of TMZ, BCNU, and/or IRI showed only moderate or no cytotoxic synergy in U251 and U87 cells. Cytotoxic synergy of combining TMZ and IRI (**A1**,**A2**), TMZ and BCNU (**B1**,**B2**), or IRI and BCNU (**C1**,**C2**) was determined in U87 cells (left) and U251 cells (right) based on their CI values. (**D1**,**D2**) Statistical analyses of best CI values for each drug combination. Bars show means (±s.e.m) of 3–4 biological replicates.

**Figure 7 ijms-25-05992-f007:**
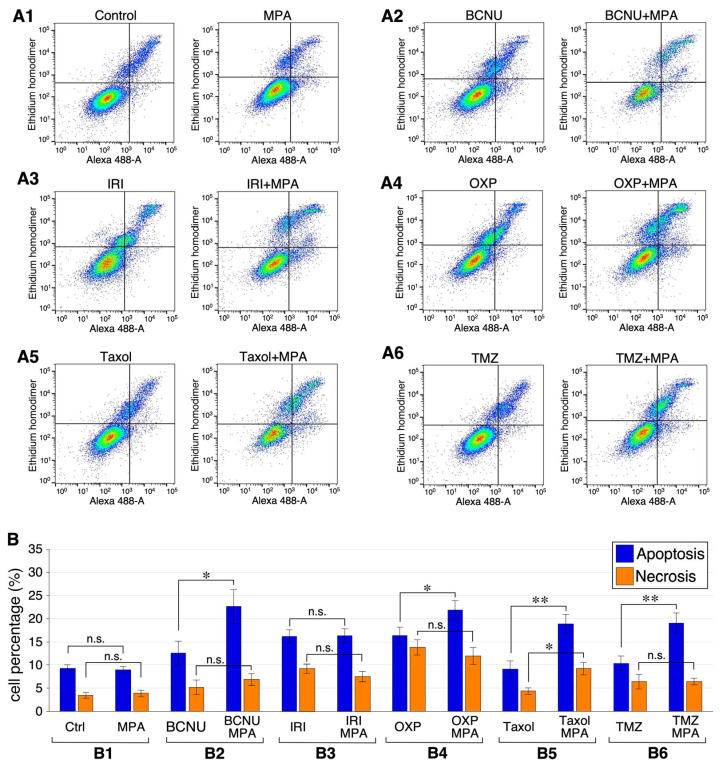
Apoptosis and necrosis of U251 cells treated with single or combined drugs. The cytotoxic effects of MPA alone (**A1**) or in combination with BCNU (**A2**), IRI (**A3**), OXP (**A4**), Taxol (**A5**), or TMZ (**A6**) were determined in U251 cells treated with single agents (1/10th of their respective IC_50_ concentrations) or MPA-combined agents (1/20th of their respective IC_50_ concentrations) for 24 h and sequentially labeled with CF488A-conjugated Annexin V (green) and EthD-III (red) for apoptosis and necrosis, respectively. (**B**) Bar graphs show means (±s.e.m.) of apoptotic (blue bars) and necrotic cell percentages (orange bars) treated with or without MPA alone (**B1**) or in the presence of BCNU (**B2**), IRI (**B3**), OXP (**B4**), Taxol (**B5**), or TMZ (**B6**) based on 4–6 biological replicates with 2–3 technical repeats each. *p*-values: *, < 0.05; **, < 0.01; n.s., no significant difference.

**Table 1 ijms-25-05992-t001:** qRT-PCR primers sequence information.

Human	Sense Primer	Anti-Sense Primer
MGMT	5′-CAGACAGGTGTTATGGAAGCTGC-3′	5′-AGAAGCCATTCCTTCACGGCCAGT-3′
TERT	5′-ACGGTGTGCACCAACATCTAC-3′	5′-GCTTTCAGGATGGAGTAGCAG-3′
EGFR	5′-AGTCAGCAGTGACTTTCTCAGC-3′	5′-GGGCACAGATGATTTTGGTCAG-3′
S6K1	5′-CAGCTCATACAAAAGCAGAACGG-3′	5′-GTAAAAGCAGGCAGTGTCTTCC-3′
P14^ART^	5′-GAGTGAGGGTTTTCGTGGTTCAC-3′	5′-CCCATCATCATGACCTGGTCTTC-3′
P16^INK4a^	5′-AGCATGGAGCCTTCGGCTGACTG-3′	5′-TGCCCATCATCATGACCTGGATC-3′
TP53	5′-GCCAAAGAAGAAACCACTGGAT-3′	5′-GCCCTTCTGTCTTGAACATGAG-3′
c-Myc	5′-AGCGACTCTGAGGAGGAACAAG-3′	5′-TGCGTAGTTGTGCTGATGTGTG-3′
STAT3	5′-AATCACGCCTTCTACAGACTGC-3′	5′-CATCCTGGAGATTCTCTACCAC-3′
TFAP2A	5′-CTCACATCACTAGTAGAGGGAG-3′	5′-AGCAGGTCGGTGAACTCTTTG-3′
CTCF	5′-AGTGTTCCATGTGCGATTACGC-3′	5′-TTCAGCTTGTATGTGTCCCTGC-3′
GABPA	5′-ACCACACTCAACATTTCGGGG-3′	5′-AGCAGATTGCACTGATGCTGG-3′
HMGN1	5′-GGCAGCAGCGAAGGATAAATC-3′	5′-TTCATCAGAGGCTGGACTCTC-3′
GAPDH	5′-GTGAAGGTCGGAGTCAAC-3′	5′-TTCTCAGCCTTGACGGTG-3′

**Table 2 ijms-25-05992-t002:** IC50 concentrations of TMZ, IRI, BCNU, MPA, OXP, and Taxol in U87 and U251 cells.

	TMZ	IRI	BCNU	MPA	OXP	Taxol
**U87**	600 μM	20 μM	200 μM	120 μM	15 μM	50 nM
**U251**	400 μM	10 μM	50 μM	120 μM	10 μM	0.3 nM

## Data Availability

Data are contained within this article.

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
