# Peer review of "IMPDH Inhibition Decreases TERT Expression and Synergizes the Cytotoxic Effect of Chemotherapeutic Agents in Glioblastoma Cells"

_ijms, 2024, doi:10.3390/ijms25115992_

Round 1

Reviewer 1 Report

Comments and Suggestions for Authors

1. the introduction section does not show well the rational work for study, the aim of study. So need more rationalization.

2. More updated references should be added to this section.

3. The synergy effects in all expermints need more detail about the aim, results presentations, and how this will be different from the other reported.

4. The conclusion does not appear written well and not reflect the novel outputs of the study at all.

5. I recommend strongly to do other malignant cell line with superproliferation than U87 and U251 cells to detect the effect of IMP dehydrogenase (IMPDH) inhibition for comparison.

Author Response

We sincerely thank the reviewers for their valid and constructive comments on our manuscript from many angles. We have addressed all the raised critiques by providing better clarifications, new data, updated citations, and justifications (marked in red). Our point-to-point responses are listed as follows.

  1. The introduction section does not show well the rational work for study, the aim of study. So need more rationalization.

Response: More details on aim/rationale are provided in the Intro’s last paragraph.

  1. More updated references should be added to this section.

Response: References are appropriately updated (also see our response to reviewer #2).

  1. The synergy effects in all experiments need more detail about the aim, results presentations, and how this will be different from the other reported.

Response: More details on the presentation of the synergy effects and how they differ from the other reported are provided in the results/methods/discussion sections.

  1. The conclusion does not appear written well and not reflect the novel outputs of the study at all.

Response: The concluding remark is revised to make sure it captures the key outputs of this paper. Please note that my concluding remark is meant to be short and concise and not to be a full paragraph.

  1. I recommend strongly to do other malignant cell line with super-proliferation than U87 and U251 cells to detect the effect of IMP dehydrogenase (IMPDH) inhibition for comparison.

Response: As interesting as it might be, testing the observed MPA effect in other malignant cell lines is not the focus of this study, which is to determine the intersection between MPA and pathways with established importance in driving the oncogenesis of adult GBM. Those selected pathways may not be important in other malignancies. Therefore, the results will not affect our conclusion, regardless of whether they are positive or negative.

Reviewer 2 Report

Comments and Suggestions for Authors

The objective was to explore the mechanism(s) underlying the efficiency of IMPDH inhibitor in glioblastoma cells. It was established that the IMPDH inhibitor MPA decreased the expression of telomerase reverse transcriptase (TERT) in both U87 and U251 cells, and the expression of O6-methyl-guanine-DNA methyltransferase (MGMT) in U251 cells. TERT downregulation was associated with a significant decrease in c-Myc in U87 cells, and an increase in p53 and CCCTC-binding factor (CTCF). MPA increased the cytotoxicity of BCNU, notably through the apoptotic pathway.

The study has several weaknesses. First, the results are not perfectly consistent between the 2 included cell lines, which limits the robustness of the conclusions. Second, there are only in vitro results (required before thinking to develop a new local-delivered treatment). Third, as discussed below, there are several methodological issues regarding the presented experiments. Fourth, the limitations are not discussed. Finally, the authors chose to explore very specific pathways. How was this choice made? How to make sure that the regulation of other major pathways (notably metabolic pathways) are not the main mechanisms underlying the efficiency of IMPDH inhibitors? Why did the authors not consider a wider approach such as RNAseq or metabolomics?

1.       In the abstract, the authors should indicate that MPA is a IMPDH inhibitor.

2.       The reference are old and need to be up-dated. For instance, reference 1 was published in 2008. There have been 2 revisions of the WHO classification of CNS tumor since this date. The authors discuss chemotherapy while they do not cite the randomized trial leaded by Stupp which represent the only consensus for GBM treatment. Reference reporting overall survival are also very old and do not correspond anymore to the reality. The same remark applies to many other citations that are also inadequate. So, the authors have to check and revise all references and not only those for which I highlighted the low relevance.

3.       Introduction: IMPDH inhibition does not date back from only a few years as stated. The reference regarding mycophenolate mofetil were published in 1996 and 2000…

4.       If the authors want to convince the readers of the relevance of their story and the potential role of MPA as a promising therapeutic agent for GBM, why do they expose the failure of MPA treatment in oncology in the introduction?

5.       Figure 1: which tumor types are included in HGG and LGG? Are these diagnosis in accordance with the WHO 2021 classification? 1B2: local, progressive and recurrent need to be defined. Recurrent only applied to diseases than can be cured (and then recur). In cases of GBM, as there is no cure, the term recur make no sense and has to be replaced by progression.

Figure 1C1: grade 3 gliomas are high-grade gliomas and not low-grade gliomas. The WHO2021 classification has to be read before leading such analysis! The authors did a lot of comparisons, sometimes overlapping as the G-CIMP phenotype for instance is linked to other studied parameters. These results lack robustness and a multivariate analysis is required to conclude. Moreover, the analyses led in the LGG subgroup suggest that IMPDH expression is a predictor of better outcome which totally compromise the relevance of targeting IMPDH, at least in LGG. Why was the survival analysis performed only in LGG and not in HGG?

6.       Figure 2: it would be nice to have a confirmation of the results at the protein level, with western blots

7.       Figure 4: p53 expression is studied. The line U251 have a p53 mutated status but not U87 cells. Was it taken into account in primer design? This point has also to be addressed in the discussion.

8.       Figure 5: it would make more sense to have tested first the synergy with TMZ as it is the only chemotherapy validated as a first line treatment for GBM.

9.       Discussion: Gliadel wafer correspond to local chemothery that is placed at the end of the surgery. This is not a post-operative but a per-operative treatment. No randomized trial has validated this strategy compared to placebo. The only validated treatment is the Stupp regimen.

10.    The limitations of the study are not discussed.

Author Response

We sincerely thank the reviewers for their valid and constructive comments on our manuscript from many angles. We have addressed all the raised critiques by providing better clarifications, new data, updated citations, and justifications (marked in red). Our point-to-point responses are listed as follows.

The objective was to explore the mechanism(s) underlying the efficiency of IMPDH inhibitor in glioblastoma cells. It was established that the IMPDH inhibitor MPA decreased the expression of telomerase reverse transcriptase (TERT) in both U87 and U251 cells, and the expression of O6-methyl-guanine-DNA methyltransferase (MGMT) in U251 cells. TERT downregulation was associated with a significant decrease in c-Myc in U87 cells, and an increase in p53 and CCCTC-binding factor (CTCF). MPA increased the cytotoxicity of BCNU, notably through the apoptotic pathway.

The study has several weaknesses. First, the results are not perfectly consistent between the 2 included cell lines, which limits the robustness of the conclusions. Second, there are only in vitro results (required before thinking to develop a new local-delivered treatment). Third, as discussed below, there are several methodological issues regarding the presented experiments. Fourth, the limitations are not discussed. Finally, the authors chose to explore very specific pathways. How was this choice made? How to make sure that the regulation of other major pathways (notably metabolic pathways) are not the main mechanisms underlying the efficiency of IMPDH inhibitors? Why did the authors not consider a wider approach such as RNAseq or metabolomics?

Response: Reviewer #2 has nicely summarized the focus and main contribution of our study, which is on the molecular mechanism(s) underlying the efficiency of IMPDH inhibitor in GBM cells. Regarding the weaknesses of our study, our responses are listed as follows.

  1. The fact that two cell lines, even of the same tumor type, don’t behave perfectly the same is a common observation in cancer research due to various factors such as genetic mutations, epigenetic changes, and cell culture conditions. This is why we conducted our studies using two cell lines instead of one and reported results that are seen in both or just one. U87 and U251 cells were chosen because they have been extensively used in GBM research. These points are included in the discussion.
  2. The focus of this paper is on the molecular mechanism underlying the effect of IMPDH inhibition (a proof-of-mechanism study), in fitting to the mission of the International Journal of Molecular Sciences. As a result, we performed our analysis using cell-based models. In vivo models will be important for its reasonable next step, that is, a translational proof-of-principle study.
  3. Methodological issues (see below)
  4. A limitation paragraph is included in the discussion.
  5. With the target-selective approach, we cannot exclude (and did not do so) the possibility that other pathways, including metabolic pathways, are also involved. Large-scale unbiased screens of genome, methylome, transcriptome, or metabolome, are valuable research strategies in making new discoveries, and each in its own right constitutes one or multiple independent studies of different scopes. This is why the scope of our study does not include high-throughput screens. Most importantly, two key weaknesses of those screen-based studies are false discoveries and a large array of targets with unclear biological and pathophysiological significance.

Methodological Issues

  1. In the abstract, the authors should indicate that MPA is a IMPDH inhibitor:

Response: The suggested change has been made.

  1. References are old and need to be up-dated. For instance, reference 1 was published in 2008. There have been 2 revisions of the WHO classification of CNS tumor since this date. The authors discuss chemotherapy while they do not cite the randomized trial leaded by Stupp which represent the only consensus for GBM treatment. Reference reporting overall survival are also very old and do not correspond anymore to the reality. The same remark applies to many other citations that are also inadequate.

Response: Suggested revisions on references have been incorporated.

  1. Introduction: IMPDH inhibition does not date back from only a few years as stated. The reference regarding mycophenolate mofetil were published in 1996 and 2000.

Response: Studies by Sintchak (1996) and Allison (2000) were cited in the original manuscript. Studies by Kofuji (2019), Shireman (2021), and Shah (2018a & 2018b) were cited to support a renewed interest in IMPDH inhibition in recently years. We don’t see an issue with those cited references.

  1. If the authors want to convince the readers of the relevance of their story and the potential role of MPA as a promising therapeutic agent for GBM, why do they expose the failure of MPA treatment in oncology in the introduction?

Response: A truthful account of what has worked and what not can only help steer the scientific community as a whole to better chances of success by avoiding repeating studies that have already been done and failed before. Our study reports a new mechanism by which MPA negatively controls GBM cell proliferation. Whether it can ultimately be applied as a therapeutic agent for GBM still requires more research, including one addressing the pharmacokinetic challenge previously reported.

  1. Figure 1: which tumor types are included in HGG and LGG? Are these diagnosis in accordance with the WHO 2021 classification? 1B2: local, progressive and recurrent need to be defined. Recurrent only applied to diseases than can be cured (and then recur). In cases of GBM, as there is no cure, the term recur make no sense and has to be replaced by progression.

Response: The NIH’s TCGA database classified grade II and III astrocytoma and oligodendroglioma as LGG (PMID: 26061751) and grade IV glioma as HGG. This classification does not completely conform to the WHO CNS5 standards (PMID: 36613601). We have made a note to this point as we discuss the results obtained from this proprietary database. TCGA classification of localized, progression, and recurrent GBMs refers to tumors that are newly diagnosed (localized), advanced/worsening over time despite initial treatment (progression), or recurrent after a period of remission following initial therapy (recurrent). The term recurrent GBM is commonly used in the literature and was used as such in the TCGA project. Since cure is uncommon for GBM, its recurrence is the rule rather than the exception. Clarifications on those terms are included in the text.

  1. Figure 1C1: grade 3 gliomas are high-grade gliomas and not low-grade gliomas. The WHO2021 classification has to be read before leading such analysis! The authors did a lot of comparisons, sometimes overlapping as the G-CIMP phenotype for instance is linked to other studied parameters. These results lack robustness and a multivariate analysis is required to conclude. Moreover, the analyses led in the LGG subgroup suggest that IMPDH expression is a predictor of better outcome which totally compromise the relevance of targeting IMPDH, at least in LGG. Why was the survival analysis performed only in LGG and not in HGG?

Response: The TCGA database defined LGG as grade II and III astrocytoma and oligodendroglioma. This point has clarified in the text. Even though higher IMPDH2 expression is associated with a better survival outcome in LGG, one cannot extrapolate this difference to HGG/GBM. In addition the noted differences in IMPDH expression are rather minor. TCGA analysis of IMPDH levels versus HGG survival was not performed because the survival information was not available in the GBM database (Affymetrix).

  1. Figure 2: it would be nice to have a confirmation of the results at the protein level, with western blots

Response: Western detection of a large protein such as TERT (120 kDa) is not as sensitive, specific, and quantitative as does qRT-PCR. For this reason, we provided data on telomere length to connect gene expression to function, which is more biologically relevant than western blot.

  1. Figure 4: p53 expression is studied. The line U251 have a p53 mutated status but not U87 cells. Was it taken into account in primer design? This point has also to be addressed in the discussion.

Response: U251 cells harbor a R273H mutation in their p53 gene, resulting in abolished DNA binding activity. This mutation occurs outside of the region detected by our qRT-PCR primers.

  1. Figure 5: it would make more sense to have tested first the synergy with TMZ as it is the only chemotherapy validated as a first line treatment for GBM.

Response: From a clinical standpoint, we understand the reviewer’s point. However, from the standpoint of exploring MPA’s mechanisms, in conjunction of the results shown in Fig. 2, it makes more sense to examine its interaction with BCNU first. Testing of the other 4 drugs, including TMZ, were done at the same time.

  1. Discussion: Gliadel wafer correspond to local chemotherapy that is placed at the end of the surgery. This is not a post-operative but a per-operative treatment. No randomized trial has validated this strategy compared to placebo. The only validated treatment is the Stupp regimen.

Response: We have revised the term post-operative to peri-operative. The reviewer’s point on the only GBM treatment validated by randomized trials is the Stupp regimen is duly noted and cited in the first paragraph of the introduction.

  1. The limitations of the study are not discussed:

Response: We have included a discussion paragraph on the limitations.

Round 2

Reviewer 1 Report

Comments and Suggestions for Authors

All were done.

Author Response

We thank Reviewer #1 for his/her approval.

Reviewer 2 Report

Comments and Suggestions for Authors

I appreciate the efforts made by the authors to take most comments into account and my global impression is that the quality of the manuscript has been improved.

I agree that U87 and U251 are widely used. However, the results would have been more robust if any other cell line was choosen in order to have more consistent results between the 2 lines. This point is now discussed, as many of the others in this revised version. References have been up-dated appropriately. To me, the elements regarding failure of MPA treatment in previous studies would be more relevant in the discussion than in the introduction. 

Some minor points remain to be adressed:

1. In figure 1, C1, C3, C4, C5, the term LGG has to be removed from the y-axis, since, as already said, grade 3 gliomas are not and have never been LGG. It is still annoying to have so many statistical tests, concerning factors dependant from each others, without any adjustment (multivariate analysis or at least Bonferroni correction).

2. Telomerase length estimation would perfectly authorize to connect gene expression and function if there were no possible TERT-Independent telomerase elongation mechanisms. If quantification of TERT proteic level (nicely performed in some studies) do not seem feasible to authors, this point has to be discussed. 

Author Response

We thank Reviewer #2 for his/her approval of our manuscript with minor revision. Our point-to-point responses are listed as follows.

  1. Response to point #1: We deleted “(LGG)” in the figure and made a note of this point in the legend.
  2. Response to point #2: The point raised on TERT+ vs. TERT- (ALT+) glioma is valid. However, U87 and U251 cells are both TERT+ cells and depend on a TERT promoter mutation (-124C>T) for long-term (30-day) proliferation (PMID: 32066906). Therefore, ALT in those cells plays more of a compensatory role than a driver role. We have included a statement on this point in the Discussion.